# The Leaching Kinetics of Iron from Titanium Gypsum in a Citric Acid Medium and Obtain Materials by Leaching Liquid

**DOI:** 10.3390/molecules28030952

**Published:** 2023-01-18

**Authors:** Yan Lin, Hongjuan Sun, Tongjiang Peng, Dingran Zhao, Xiyue Zhang

**Affiliations:** 1Key Laboratory of Solid Waste Treatment and Resource Recycle, Ministry of Education, Mianyang 621010, China; 2Institute of Mineral Materials and Application, Southwest University of Science and Technology, Mianyang 621010, China; 3Analytical and Testing Center, Southwest University of Science and Technology, Mianyang 621010, China

**Keywords:** titanium gypsum, calcium oxalate, ferrous oxalate, ferric hydroxide, iron removal rate

## Abstract

In this study, the effect of citric acid on iron leaching from titanium gypsum (TiG) was systematically investigated. The conditions for the leaching of valuable metals were optimized while varying such parameters as the leaching time, citric acid mass fraction, leaching temperature, and the liquid–solid ratio. It was found that under the conditions of a citric acid mass fraction of 10%, at a 80 °C leaching temperature, a leaching duration of 80–90 min and a liquid–solid ratio of 8, the whiteness of titanium gypsum (TiG) increased from 8.1 to 36.5, and the leaching efficiencies of iron reached 84.37%. The kinetic analysis indicated that the leaching process of iron from TiG was controlled by the reaction product layer from 0–20 min, while the leaching process of iron from TiG was controlled by internal diffusion from 20–90 min. The apparent activation energy of the leaching reactions was 33.91 kJ/mol and 16.59 kJ/mol, respectively. High-value-added calcium oxalate and ferrous oxalate were prepared from the calcium and iron in the filtrate of the oxalic acid extraction. The leaching liquid could be recycled, which will provide a new way to utilize titanium gypsum.

## 1. Introduction

Titanium gypsum (TiG) is obtained during the production of titanium dioxide, by the sulfuric acid method, and is a kind of solid waste largely composed of calcium sulfate dihydrate. The production of 1 ton of titanium dioxide is necessary to generate about 6–10 tons of TiG [1,2,3]. China emits 30 million tons of TiG every year and Huntsman produces 925,000 tons of TiG every year worldwide [4,5]. Because of the small particles of calcium sulfate dihydrate, high free-water content, high viscosity, and high ferric hydroxide content in TiG, the large-scale resource utilization of titanium gypsum is limited by a long stacking treatment time, pollution to arable land, water, and air, seriously restricting the healthy and green development of the titanium dioxide industry.

TiG contains iron hydroxide, a small amount of sulfate, aluminum hydroxide, and other impurities, which have many adverse effects on the use of TiG, such as moisture absorption, frost return, whiteness, and mechanical properties of the products [6,7]. Azdarpour, et al. [8] studied the effect of alkali and acidic solutions on the leaching of calcium and iron from TiG as a raw material. The results showed that the alkali solutions (NaOH, KOH) could not extract large amounts of calcium or iron, while the acidic solutions (H_2_SO_4_, HCl, and HNO_3_) had high extraction efficiency, and the extraction efficiency of H_2_SO_4_ was higher than that of HCl and HNO_3_. PENG Xiaoqian et al. [9] effectively removed iron from red gypsum residue under hydrothermal conditions by controlling the phase transformation of the gypsum and the morphology of iron. The results showed that under optimized treatment conditions of a liquid-to-solid ratio of 10, leaching agent 1.5 M HCl and heating at 140 °C for 6 h, the iron removal rate was above 99%. Jiang Meixue et al. [10] used TiG as the raw material to remove iron hydroxide from titanium gypsum by sulfuric acid leaching. The results showed that sulfuric acid leaching effectively removed ferric hydroxide from titanium gypsum, with a removal rate reaching 93.14% under the best technological conditions, and the whiteness increased from 8.1 to 54.4.

At present, there are relatively few experimental studies on the removal of iron and impurities from TiG, mainly focusing on acid leaching because of its good and fast removal of iron. Although sulfuric acid, hydrochloric acid, and other strong acids, good effect, they can easily corrode equipment. Therefore, it is of great experimental significance to find mild and efficient iron and impurity-removing agents. As a weak acid, citric acid is not only a retarder of gypsum cement [11,12], but also an effective solvent for removing iron. Citric acid iron removal has been studied in clay minerals [13,14,15], but titanium gypsum iron removal has not yet been reported. This method can not only effectively separate impurities, such as ferric hydroxide and gypsum, and improve the whiteness and purity of gypsum products after iron removal, but also prepare high-value-added calcium oxalate and ferrous oxalate from the leaching liquid. Calcium oxalate is an inorganic non-metallic material with high-added-value. Because of its excellent optical properties, it can be used as a pigment and filler in the paper-making industry, the ceramic-glazing industry, etc. It can also be used as a carrier in the separation of rare metals. Ferrous oxalate is also an inorganic non-metallic material with high-added-value. It is used as a photographic developer, in the pharmaceutical industry, and as a raw material for lithium iron phosphate, a cathode material for batteries. This is a significant new and worthy topic of study, which lays the foundations for the large-scale comprehensive utilization of TiG.

## 2. Experimental Part

### 2.1. Raw Samples and Reagents

TiG was purchased from Sichuan Panzhihua waste residue produced by limestone gypsum wet-processing of titanium dioxide using sulfuric acid. TiY had a reddish color. TiG was screened using a standard inspection sieve, dried to constant weight at 60 °C, sealed, and stored in a dryer for later use. Citric acid monohydrate (AR, C_6_H_8_O_7_·H_2_O ≥ 99.5%), oxalic acid dihydrate (AR, H_2_C_2_O_4_·2H_2_O ≥ 99.5%), reduced iron powder (AR, Fe ≥ 98%), and hydrogen peroxide (AR, H_2_O_2_ ≥ 30%) were purchased from Kelong Chemical Reagent Co., Ltd., Chengdu, China. We used laboratory homemade deionized water, resistivity > 18.2 MΩ·cm.

### 2.2. Scheme and Procedure

The effects of the citric acid mass fraction, reaction temperature, liquid–solid ratio, and reaction time on the iron removal rate of titanium gypsum were investigated by a single-factor experiment, and the optimal technological conditions were determined. The specific experimental conditions for the efficient iron removal by citric acid leaching is shown in Table 1. The experimental procedure for the recycling the secondary leaching liquid treatment is as follows: (1) 50 mL of citric acid leaching liquid was added 5 mL of hydrogen peroxide in a 20 °C water bath, and allowed to react for 10 min. (2) 4.5 g of dihydrate oxalic acid was added to the mixture and allowed to react for 50 min, calcium oxalate and filtrate were obtained by filtration. (3) The filtrate was added with 1 g of reductive iron powder, reacted at 60 °C for 20 min. (4) Then, the filtrate was reacted at 20 °C for 180 min to obtain ferrous oxalate. The excess iron powder was removed with a magnet and the filtrate was reused.

2.5 g of titanium gypsum was placed in a 50 mL tapered flask. To this a certain volume and mass fraction of citric acid aqueous solution was added and mixed well. The mixture was allowed to soak in a constant temperature oscillator from a water bath at a rotational speed of 150 r/min at a set reaction temperature for a period of time. Subsequently, the mixture was removed and filtered while still hot. The filter cake was then fully washed and dried at 60 °C to obtain the acid leaching gypsum products. The number of samples is shown in Table 1.

### 2.3. Data Analysis

The phase composition of the produce was studied using an X-ray diffractometer (XRD, Ultima IV, Akishima, Japan) on randomly oriented specimens. Measurements were carried out with a scanning speed of 0.02 deg/s, an X-ray tube voltage of 40 kV, and a tube current of 40 mA. The Panaco Axios X-ray fluorescence spectrometer (XRF, PANalytical Corporation, Holland) was used to determine the chemical composition of the sample (wavelength dispersion type) after iron removal in the original sample and the op-timal solution. The test conditions were ceramic X-ray tube (Rh target) with a maximum power of 2.4 kW. Scanning electron microscopy (SEM) of the samples was performed using a Zeiss Ultra 55 microanalyzer (Zeiss Instrument Company, Germany) with an accelerat-ing voltage of 15 kV. The sample whiteness test was performed with SBDY-1P, and the ref-erence whiteboard was R457:84. we have used CIT-3000F building materials radioactive detection analyzer (Sichuan Xinxianda Measurement and Control Technology Co., Ltd., China) to test TiG sample, according to the NATIONAL standard GB6656-2010. Differen-tial thermal analysis and thermal-gravimetric analysis (TG-DTA,) of the products was an-alyzed using the thermal analyzer method with the SDTQ600 analyzer (TA Instrument Company, America)using an air atmosphere heating rate of 10 °C/min. The dissolution amount of TiG was weighed with an electronic weighing instrument. According to the GBT3884.15-2014 potassium dichromate titration method of total iron, determine the con-tent of TFe in the original sample and the sample after acid leaching.
$$\mathrm{Iron}\;\mathrm{Removal}\;\mathrm{Rate}\;\left(\%\right)=\frac{\begin{array}{c} Percentage\;of\;TiG’s\;TFe-\\ Percentage\;of\;Sample\;after\;acid\;leaching’s\;TFe \end{array}}{\mathrm{Percentage}\;\mathrm{of}\;\mathrm{TiG}’s\;\mathrm{TFe}}$$

### 2.4. Theoretical Basis

After acid leaching, the citric acid of the sample not only has the characteristics of an acid but also can react with the calcite in the TiG to neutralize the iron-containing compounds. This makes the reaction products dissolved in water diffuse more easily into the solution and accelerate the dissolution process. In this study, the following chemical Reactions (1)–(3) occurred during the leaching of iron compounds from the TiG in citric acid. Chemical Reactions (4)–(7) occurred in the preparation of calcium oxalate and ferrous oxalate by the leaching liquid treatment.
Fe(OH)_3_ + 3H^+^ = Fe^3+^ + 3H_2_O(1)
Fe(OH)_2_ + 2H^+^ = Fe^2+^ + 2H_2_O(2)
CaCO_3_ + 2H^+^ = Ca^2+^ + CO_2_↑ + H_2_O(3)
2Fe^2+^ + 2H^+^ + H_2_O_2_ = 2Fe^3+^ + 2H_2_O(4)
Ca^2+^ + C_2_O_4_^2−^ + 2H_2_O = CaC_2_O_4_·2H_2_O↓(5)
2Fe^3+^ + Fe = 3Fe^2+^
(6)
Fe^2+^ + C_2_O_4_^2−^ + 2H_2_O = FeC_2_O_4_·2H_2_O↓(7)

## 3. Results and Discussion

### 3.1. Analysis of the Titanium Gypsum Raw Samples

The chemical composition of TiG is given in Table 2. Table 2 shows the results of the XRF analysis of the raw material samples. It can be seen that the main chemical composition of the titanium gypsum was SO_3_ 39.13%, CaO 38.48%, TFe_2_O_3_ 11.18%, TiO_2_ 2.52%, SiO_2_ 4.05%, and MgO 2.08%. Containing small amounts of Al_2_O_3_, Na_2_O, MnO, and Cl and trace amounts of Na_2_O, As_2_O_3_, SrO, ZnO, ZrO_2_ and K_2_O, the losses from combustion were up to 21.6%, and 6.44% for TFe. The TFe in the TiG sample titrate by the above titration method was 6.27%, with a difference of 0.17%. The pH value of the samples was 8.18, so it is speculated that the iron in the samples was in the form of iron hydroxide.

Figure 1a shows the TiG XRD pattern, where only two phases of the TiG were gypsum and calcite. The main characteristic diffraction peaks of TiG are d_020_ = 7.6346, d_021_ = 4.2896, d_041_ = 3.0674, d_-221_ = 2.8735 (1 Å = 0.1 nm). The gypsum peak at 3.067 nm covers the calcite peak at 0.304 nm, and the observed gypsum peak at 3.067 nm has a large peak width, which is consistent with the literature [16]. After the addition of citric acid in the later stage, a large number of bubbles were produced, which were tested with a flame-smoking stick. These bubbles extinguished the flame, proving that the gas produced was CO_2_. Calcite was identified in the TiG sample. Figure 1b shows the raw material sample of TiG under an electron microscope. It can be seen from the figure that most of the gypsum particles were very small, with only a small amount of diamond-shaped gypsum and block-shaped calcite particles. According to the chemical composition analysis of the TiG raw materials in Table 2, although there were many components of iron hydroxide in the TiG, the characteristic diffraction peaks of related minerals was not detected in the XRD pattern, indicating that the iron hydroxide components of the titanium gypsum mainly existed in an amorphous state, consistent with the results in the literature [17,18,19]. Figure 1c shows the optical microscope map of the TiG material samples; as seen from the diagram, most of the iron compounds were distributed on the surface of the titanium gypsum. Iron compounds attached on the TiG surface hinders the growth of gesso, resulting in iron compounds being distributed on the gypsum crystal nucleus, thus leading to an irregular shape of the TiG and small gypsum particles. The radioactive test results are given in Table 3. The TiG sample is classified as Class A material, according to the NATIONAL standard GB6656-2010.

### 3.2. Total Iron Leaching Rate and Phase Change of the Acid Leaching Products

#### 3.2.1. Effect of the Citric Acid Mass Fraction

Figure 2a shows the variation in the amount of TiG dissolution and the iron removal rate with the mass fraction of citric acid monohydrate. The analysis shows that the amount of titanium gypsum dissolution increased with increasing mass fraction of citric acid monohydrate. When the mass fraction of citric acid monohydrate was 0–12.5%, the iron removal rate increased from 9.09% to 82.46%. When the mass fraction of citric acid monohydrate was 10%, the iron removal rate was 82.14%. When the mass fraction of citric acid monohydrate was greater than 10%, the iron removal rate slowly increased with the mass fraction of citric acid monohydrate.

Figure 2b shows the XRD patterns of the acid leaching products with different mass fractions of citric acid monohydrate. The analysis shows that when the mass fraction of citric acid monohydrate was 0–12.5%, the main phase of the product after acid leaching was gypsum dihydrate. When the mass fraction of citric acid monohydrate ranged from 1.0% to 7.5%, a small amount of crystalline and amorphous calcium citrate tetrahydrate was found. It can be seen from the figure that when the mass fraction of citric acid monohydrate exceeded 5%, the peak width of gypsum at 3.067 nm is still large, so it can be judged that the sample after acid leaching still has calcite. When the mass fraction of citric acid monohydrate exceeded 5%, the maximum width of gypsum at 3.067 nm decreased and did not change, thus it can be concluded that the calcite reaction was complete.

The increased value of citric acid monohydrate in a low-quality fraction as a result of the acid and alkali neutralization reaction of ferrous iron or citric acid in the solution leads to an increased iron removal rate; as a result of a small concentration of the reagent being present on the surface of the iron hydroxide TiG, the leaching of iron depends mainly on the initial concentration of citric acid, the higher the initial concentration, the greater the leaching rate. However, the content of iron compounds in the system was constant, so when the mass fraction of citric acid monohydrate increased to a certain critical point, that is, when the mass fraction of citric acid monohydrate was sufficient to react with the iron compounds in the system, the total iron removal rate did not significantly change. At 1–2.5%, the amount of titanium gypsum dissolution was slightly reduced due to the production of calcium citrate tetrachronic acid, but the neutralization reaction of iron compounds and citric acid was not affected, so the removal rate of total iron still increased.

#### 3.2.2. Effect of Reaction Temperature

Figure 3a shows the variation in the amount of titanium gypsum dissolution and the iron removal rate with the reaction temperature. The analysis shows that the amount of titanium gypsum dissolution increased with increasing reaction temperature. When the reaction temperature was between 25 °C and 95 °C, the iron removal rate increased from 22.49% to 82.78%, and the dissolution rate of titanium gypsum did not change significantly between 80–95 °C. When the reaction temperature was 80 °C, the iron removal rate was 82.14%. When the reaction temperature was higher than 80 °C, the iron removal rate increased slowly with the mass fraction of citric acid monohydrate. When the reaction temperature was 95 °C, the iron removal rate reached 82.78%. As the temperature rose, the equilibrium favored a shift in the direction of heat absorption. The temperature increase was beneficial to overcome the interaction between the substance molecules, so that the substance molecules leave the surface of the substance into the solvent process. Therefore, the increase in temperature was conducive to the dissolution of soluble impurities. Because of the acid and alkali neutralization reaction, citric acid iron enters into the solution, and the increase in temperature increased the diffusion coefficient and rate constant, increasing the removal rate of iron. Because the iron compound content was constant, when the temperature increased to a certain critical point, the removal rate of total iron in the titanium gypsum changed.

Figure 3b shows the XRD patterns of the acid leaching products at different reaction temperatures. The analysis shows that when the reaction temperature was between 25–95 °C, gypsum was the only phase of the product after acid leaching. With increasing temperature, the crystallinity of gypsum tended to decrease. The article by Lanzón Marcos [20] showed that the side effect of citric acid as a highly effective retarder for gypsum is to reduce the strength of gypsum.

#### 3.2.3. Effect of the Liquid–Solid Ratio

Figure 4a shows the variation in the amount of titanium gypsum dissolution and the iron removal rate with differing ratios of liquid to solid. The analysis shows that the dissolution amount of titanium gypsum increased slightly with increasing liquid–solid ratio. When the liquid–solid ratio exceeded 8, the dissolution amount of titanium gypsum did not change significantly. When the liquid–solid ratio was 6–14, the removal rate of TFe increased from 74.32% to 83.98%. When the liquid–solid ratio increased to 8, the removal rate of TFe increased to 81.98%, and the removal rate of TFe remained unchanged. As ferric citrate, formed by the acid–base neutralization reaction, entered the solution, the removal rate of iron increased, but the content of iron compounds in the system was constant. Therefore, when the liquid–solid ratio continued to increase to a certain critical point, the total iron removal rate in titanium gypsum did not change significantly.

Figure 4b shows the XRD patterns of the acid leaching products with different liquid–solid ratios. The analysis shows that when the liquid–solid ratio was 6–14, only the gypsum phase peak was found in the products after acid leaching. The peak strength of the gypsum phase decreased with an increase in the liquid–solid ratio. The article by Jiahui, P. [21] showed that due to the preferred adsorption of citric acid on the face [111], the growth of the C axis is strongly inhibited, consequently leading to the transformation of the dihydrate crystals.

#### 3.2.4. Effect of Reaction Time

Figure 5a shows the variation in the amount of TiG dissolution and the iron removal rate with reaction time. The analysis shows that the dissolution amount of TiG increased rapidly and slightly with increasing reaction time. When the reaction time was 20 min, the dissolution rate of TiG was 0.8814 g/25 mL. The amount of TiG dissolution did not change significantly after 20 min. When the reaction time was 20–90 min, the TFe removal rate increased from 72.89% to 84.37%. When the reaction time was 60 min, the removal rate of TFe was 81.98%. When the reaction time was 90 min, the removal rate of TFe increased to 84.37%. As the change is not obvious, the reaction time range should therefore be optimized after at least 60–90 min.

Figure 5b shows the XRD patterns of the acid leaching products at different reaction times. The analysis shows that when the reaction time was 10–90 min, only gypsum was produced after acid leaching, thus the reaction time did not affect the TiG phase. The maximum strength of the phase increased with increasing reaction time. The maximum strength of the phase did not change significantly with a reaction time of 10–60 min, but between 60–90 min the maximum strength of the phase changed significantly. Therefore, we suggest that the dissolution time should be above 90 min.

The general leaching process is divided into three diffusion chemical reaction processes; therefore, for the leaching of titanium-containing iron compounds, with increasing reaction time due to acid and alkali neutralization reactions of citric acid iron and the increase in iron removal rate, the iron compound content is constant in the system, so when the reaction time increases to a critical point, even for prolonged reaction times, the removal rate of total iron from titanium gypsum cannot be significantly changed.

#### 3.2.5. Effect of the Number of Reaction Cycles

Table 4 displays the sample chemical composition analysis with different cycles numbers. As can be seen from the table, the TFe removed from the titanium gypsum with cycles 1, 2 and 3 was 0.86%, 1.12% and 5.15%, respectively. The pH of a 10% mass fraction measured during the experiment was 1.77. The pH of the leached liquid after the first cycle was 2.97. The pH of the leached liquid after the second cycle was 4.16. It is easy to conclude that the lower the pH, the more favorable the leaching of iron from TiG. The iron removal rates were 86.73%, 75.3% and 20.52% by semi-quantitative chemical composition analysis. Compared with Figure 5a, it can be seen that the method of calculating the iron removal rate by semi-quantitative chemical composition analysis was larger (the deviation was less than 5%) for the titration of total iron than the potassium dichromate titration method of total iron. As can be seen from Table 4, the calcium and sulfur elements in the first two cycles were greater than 90%. Under the optimized process conditions, citric acid not only had an obvious leaching effect on iron, but also on other elements, especially sodium, magnesium, manganese, zinc, chlorine, arsenic and other elements.

Figure 6 shows the XRD patterns of samples under different numbers of cycles. The analysis shows that when the number of cycles was 1–3, only gypsum was produced after acid leaching, so the number of cycles did not affect the TiG phase. Compared with the XRD pattern of the original titanium gypsum in Figure 1, the gypsum crystallization strength of the citric acid-treated gypsum sample increased significantly. From the crystallinity level, the citric acid leaching process had a significant effect on the crystallization strength of gypsum. The side effect of reducing the strength of gypsum with citric acid as a highly effective retarder may be resolved.

#### 3.2.6. Leaching Kinetics of TFe

##### Effect of Temperature and Time

Figure 7a shows the influence of different temperatures and times on the amount of titanium gypsum dissolution. As can be seen from the figure, the amount of titanium gypsum dissolution increased significantly at 2–20 min. At 95 °C, the amount of titanium gypsum dissolution reached a maximum of 0.9372 g/25 mL; while at 20–90 min, the amount of dissolution increased gently; at 40 °C, the maximum increase amount was 0.1379 g/25 mL. The dissolution of impurities in titanium gypsum can be divided into two stages. Figure 7b shows the influence of the TFe removal rate at different temperatures and times. As can be seen from the figure, iron removal was concentrated mainly in the first 20 min. At 95 °C, the TFe removal rate reached its highest at 77.51%, while the TFe removal rate increased gently from 20 to 90 min, and the maximum increase in the TFe removal rate was less than 20%. This kinetic curve was consistent with the kinetic characteristics of the reaction in the multiphase liquid–solid region [22,23], that is, the first stage is the accelerated reaction stage, and the second stage is the gentle completion stage.

##### The Model of Leaching Kinetics

The study of leaching kinetics focuses on determining the relationship between the reaction rate and basic parameters. When the concentration of one reactant is much too high compared with that of another reactant, the concentration of this reactant can be regarded as a certain value to determine the kinetic equation. According to the reaction principle, this reaction process belongs to the liquid–solid phase reaction process. During the whole leaching process, the concentration of reactant body is basically constant, and the model of shrinking the unreacted core can be selected to describe it.

When the reaction is controlled by chemical reaction [24,25,26], the equation is expressed as follows:(8)$$1-(1-x{)}^{\frac{1}{3}}=kt$$

When the reaction is controlled by the reaction product layer [27,28,29,30], the equation is expressed as follows.
(9)$$1-3{\left(1-x\right)}^{\frac{2}{3}}+2\left(1-x\right)=kt$$

When the reaction is controlled by internal diffusion [31,32,33,34], the equation is expressed as follows.
(10)$$1-{\left(1-x\right)}^{\frac{2}{3}}-\frac{2}{3}x=kt$$

The influence of temperature on the leaching rate is expressed mainly as the rate constant *k*, whose relationship with absolute temperature follows the Arrhenius formula [35]:(11)$$k=k_{0}\exp\left(-\frac{E_{a}}{\mathrm{RT}}\right)$$

The logarithm of both sides of Equation (11) is obtained
(12)$$\ln k=\ln k_{0}-\frac{E_{a}}{\mathrm{RT}}$$

*K*_0_ is the frequency factor; *E_a_* is the apparent activation energy, kJ/mol; R is the constant molar gas, 8.314; and T is the temperature in Kelvin, K.

According to the shrinkage unreacted core model and the different reaction conditions, the chemical kinetics of the leaching of iron compounds in a ferric citrate solution can be divided into chemical reaction control, diffusion control, and product layer diffusion control. It is speculated that the leaching reaction process of iron compounds is composed of the chemical reaction steps at the liquid–solid interface and a diffusion step of the hydrated citrate product generated through the reaction coating on the unreacted layer. The whole reaction process can be controlled by the chemical reaction steps or the diffusion step of the reaction product layer, or other steps. The experimental results of 2–20 min and 20–90 min were adjusted by chemical reaction control, reaction product layer control, and internal diffusion control, respectively, as shown in Figure 8a–f. As can be seen from the figure, iron leaching results fitted better with the reaction product layer control and internal diffusion control.

To further verify this, ln *k* and 1/T were fitted according to the data in Figure 7, and the relationship curve between ln *k* and 1/T was obtained, as shown in Figure 9.

The reaction activation energy *E_a_* was calculated according to the slope in Figure 9. Table 5 shows the apparent activation energy *E_a_* and the frequency factor *K*_0_. From the table, it can be seen that the first 2–20 min conforms to the control of the reaction product layer, and 20–90 min conforms to the internal diffusion control. The kinetic model for the leaching of iron compounds from titanium gypsum in citric acid is as follows:

Within 2 to 20 min
(13)$$1-3{\left(1-x\right)}^{\frac{2}{3}}+2\left(1-x\right)=1305.315\exp\left(-\frac{33.91}{\mathrm{RT}}\right)t$$

Within 20 to 90 min
(14)$$1-{\left(1-x\right)}^{\frac{2}{3}}-\frac{2}{3}x=0.238\exp\left(-\frac{16.59}{\mathrm{RT}}\right)t$$

The value of *E*_a_ under the control of the chemical reaction was lower than under the control of internal diffusion within 2–20 min, indicating that the leaching process under the control of internal diffusion requires a higher external energy. A higher activation energy usually means that the leaching process may require a higher solution temperature through heating, or that a highly reactive leach agent is required. According to the above conclusion, we can concluded that in the first 20 min, citric acid is deep within the iron hydroxide particles along the channel space of diffusion at the same time, reactant molecules react on the whole wall, the ferric citrate complex, the inside of the reactants deposit on the surface of the particles forming a loose and porous inert layer, and the diffusion of the molecules causes a certain amount of resistance. The rate of diffusion is almost equal to the rate of the chemical reaction. The higher the activation energy, the lower the reaction rate, and the higher the diffusion resistance. Furthermore, the apparent activation energy of the diffusion control of the product layer should be between the chemical control and internal diffusion control (20–40 kJ·mol^−1^). After 20 min, with an increase in time, the smaller particles in the nuclear area, citric acid and iron hydroxide spread along the channel space simultaneously, diffusing within the nuclear reaction completely, not forming the product layer, or the sediments of the products. Diffusion control is caused by chemical reactions occurring too quickly for diffusion to supply enough molecules to the reaction. The rate control step changes from diffusion control through the product layer to diffusion control at the surface. This process is controlled by the internal diffusion effect, affected by temperature, the higher the temperature, the greater the reaction rate. 

### 3.3. Extraction of Calcium and Iron from the Leaching Liquid

Figure 10a shows the Raman spectra of calcium oxalate and ferrous oxalate produced under experimental conditions. It can be seen from the figure that the wave number is 100 cm^−1^–300 cm^−1^, and there are strong Raman vibration peaks between. The peak values of calcium oxalate are 259 cm^−1^, 220 cm^−1^, 188 cm^−1^ and 162 cm^−1^, which can be derived from the stretching and bending vibrations of Ca−O. Ferric oxalate peak values are 293 cm^−1^, 246 cm^−1^, and 203 cm^−1^, due to the stretching and bending vibrations of Fe−O. Calcium oxalate has weak peaks at 596 cm^−1^ (broadened, due to the water vibration mode) and 505 cm^−1^ (possibly due to the symmetrical O−C=O bending mode), and ferrous oxalate has vibration peaks at 582 cm^−1^ and 518 cm^−1^ (possibly due to the symmetrical O−C=O bending mode). Raman frequency peaks near 900 cm^−1^ are derived from C−C stretching mode, calcium oxalate at 909 cm^−1^ and ferrous oxalate at 913 cm^−1^. There is a vibration stretching mode V (C−O) in the wave number range 1456 cm^−1^ to 1473 cm^−1^. Both calcium oxalate dihydrate and ferrous oxalate have a vibration peak at 1468 cm^−1^, and a vibration peak of the symmetrical stretching mode appears near the low wave number. The vibration peak of calcium oxalate dihydrate is 1411 cm^−1^, and that of ferrous oxalate is 1450 cm^−1^, which can be derived from the B2g mode (O−C=O vibrations). The vibration peaks of calcium oxalate dihydrate at 1628 cm^−1^ and 1737 cm^−1^, and that of ferrous oxalate at 1555 cm^−1^ and 1707 cm^−1^, may be caused by the stretching mode of V (C=O). We observed that the OH vibration peaks of the two oxalates were significantly different. The vibration peak of calcium oxalate dihydrate was 3467 cm^−1^, and the vibration peak of ferrous oxalate was 3315 cm^−1^.

Figure 10b shows the XRD patterns of calcium oxalate and ferrous oxalate. The main characteristic diffraction peaks of calcium oxalate are *d*_200_ = 6.1872 Å, *d*_121_ = 4.4368 Å, and *d*_141_ = 2.7836 Å, and the cell parameters of calcium oxalate are a = 12.371, b = 12.371, c = 7.357, α = 90 °C, β = 90 °C, and γ = 90 °C, with a cell volume of 1125.9 Å^3^. The sample was not much different from the cell parameters of the standard PDF card (75–1314). It can be determined that calcium oxalate has symmetry type D_3_, and its space group is expressed as $I_{4/m}$, Z = 8. The strongest peak d values of ferrous oxalate are *d*_200_ = 4.8237 Å, *d*_-202_ = 4.8237 Å, and *d*_-402_ = 3.0377 Å. The sample was not much different from the cell parameters of the standard PDF card (89–7120).

## 4. Conclusions

In this study, the leaching behavior of citric acid in leaching iron from titanium gypsum was studied. The conclusions from this work are summarized below:

(1)The results of the leaching experiment showed that citric acid can extract a large amount of iron from titanium gypsum, and the leaching conditions were optimized as follows: At 80 °C, the mass fraction of citric acid was 10%, the leaching time was 80–90 min, and the liquid–solid ratio was 8. The iron removal rate of TFe reached 84.37%, broadening the range of applications for titanium gypsum.(2)The iron removal rate of TFe calculated by XRF semiquantitative chemical composition analysis was 86.65%. The iron removal rate of TFe calculated by XRF semiquantitative chemical composition analysis was larger than the iron removal rate of TFe titrated by potassium dichromate titration (the deviation was less than 5%), and the sum of calcium and sulfur elements was more than 90%. The quality of titanium gypsum was improved.(3)Kinetic analysis showed that the leaching kinetics of TiG TFe in citric acid was controlled by diffusion of the product layer in the first 20 min and then by internal diffusion afterwards.(4)Under certain conditions, calcium oxalate and ferrous oxalate were prepared with high-added-value from a citric acid leaching solution. This provides the basis for the full utilization of valuable compositions of titanium-gypsum as a resource.

## Figures and Tables

**Figure 1 molecules-28-00952-f001:**
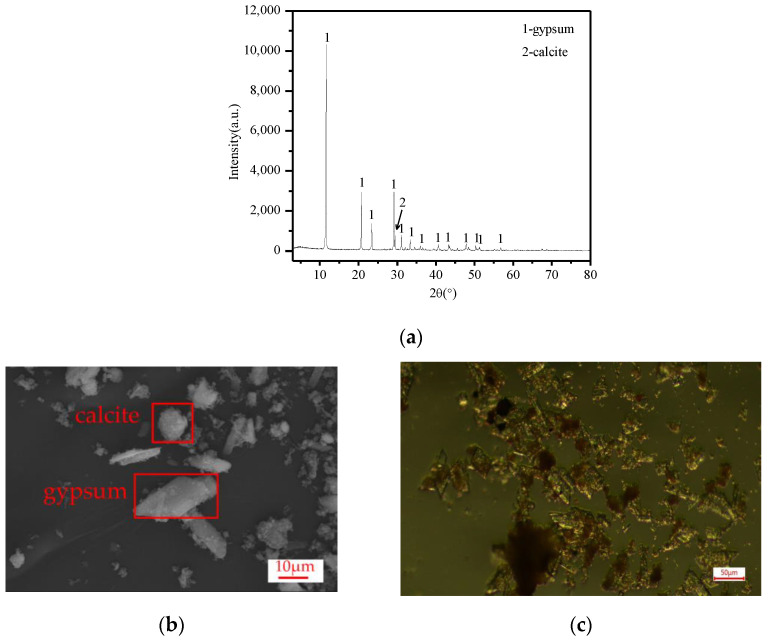
(**a**) XRD patterns of the TiG raw materials, (**b**) SEM spectra of the TiG raw material samples, and (**c**) optical microscope atlas of the TiG raw samples.

**Figure 2 molecules-28-00952-f002:**
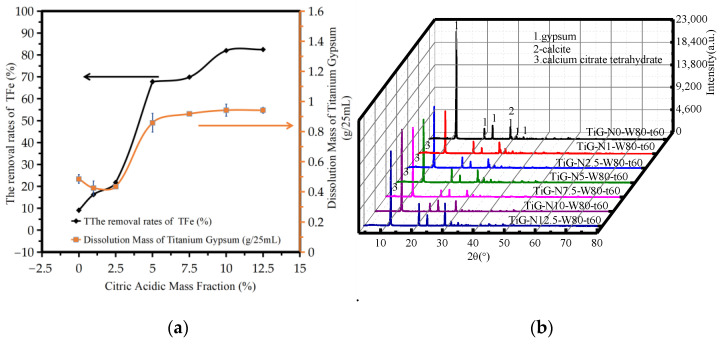
Influence of the mass fraction of citric acid monohydrate. (**a**) The dissolution amount of TiG and the removal rate of total iron. (**b**) The XRD patterns of the acid leaching products.

**Figure 3 molecules-28-00952-f003:**
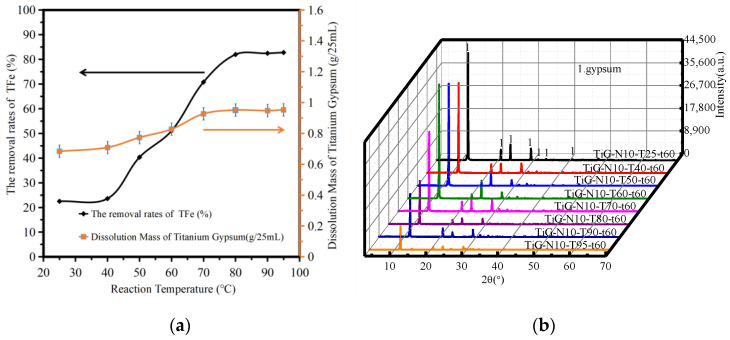
The influence of reaction temperature. (**a**) The dissolution amount of TiG and the removal rate of total iron. (**b**) The XRD patterns of the acid leaching products.

**Figure 4 molecules-28-00952-f004:**
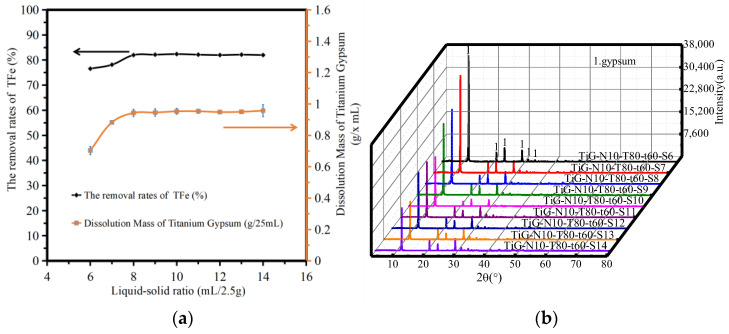
The influence of reaction temperature. (**a**) The dissolution amount of TiG and the removal rate of total iron. (**b**) The XRD patterns of the acid leaching products.

**Figure 5 molecules-28-00952-f005:**
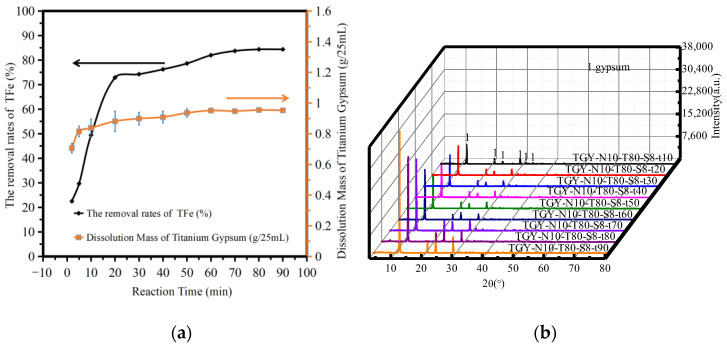
The influence of reaction time. (**a**) The dissolution amount of TiG and the removal rate of total iron. (**b**) The XRD patterns of the acid leaching products.

**Figure 6 molecules-28-00952-f006:**
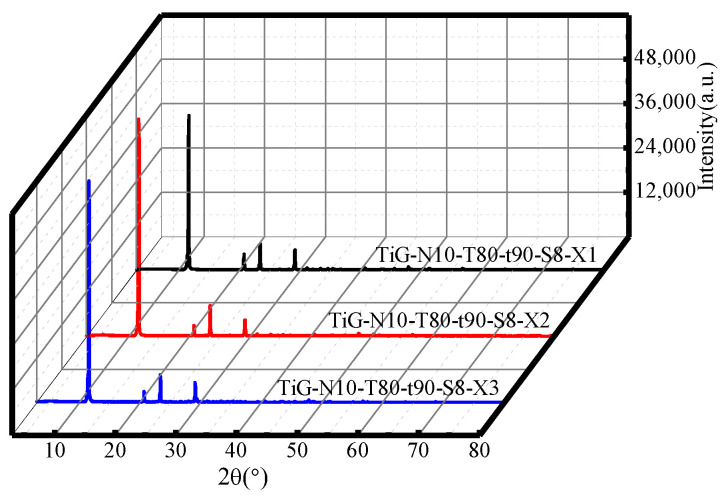
XRD patterns of the acid leaching products with different numbers of cycles.

**Figure 7 molecules-28-00952-f007:**
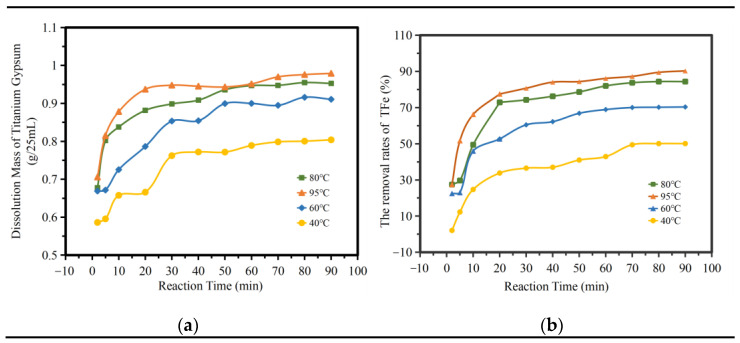
The influence of reaction time. (**a**) The dissolution amount of TiG. (**b**) the removal rates of total iron.

**Figure 8 molecules-28-00952-f008:**
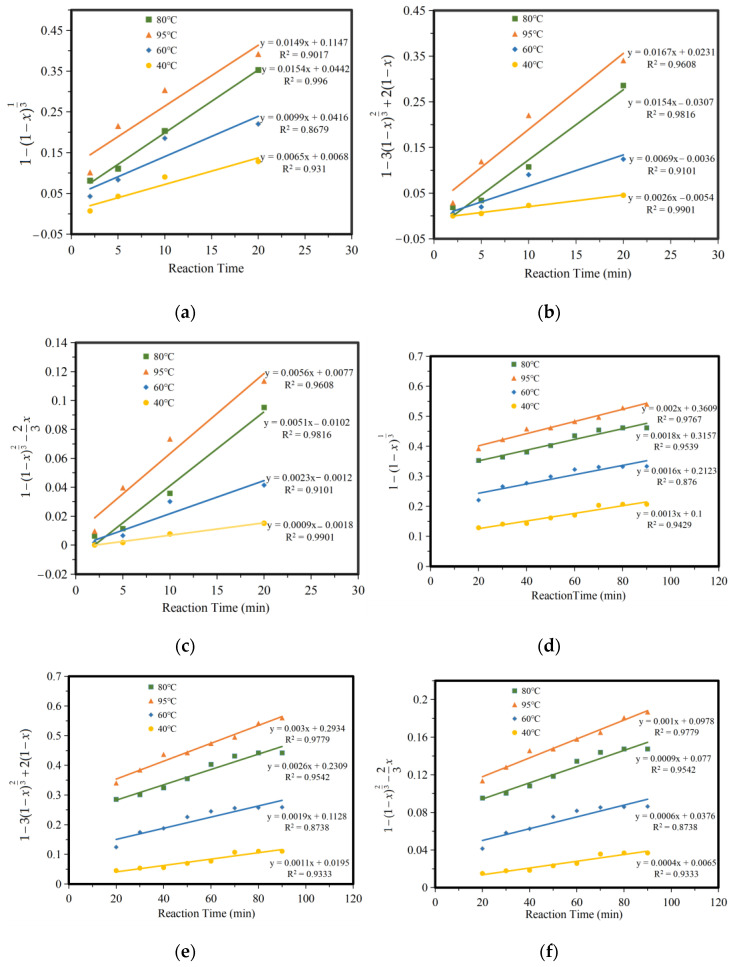
Fitting of iron leaching control. (**a**) 2–20 min chemical reaction control, (**b**) 2–20 min product layer diffusion control, (**c**) 2–20 min membrane diffusion control, (**d**) 20–90 min chemical reaction control, (**e**) 20–90 min product layer diffusion control, (**f**) 20–90 min membrane diffusion control.

**Figure 9 molecules-28-00952-f009:**
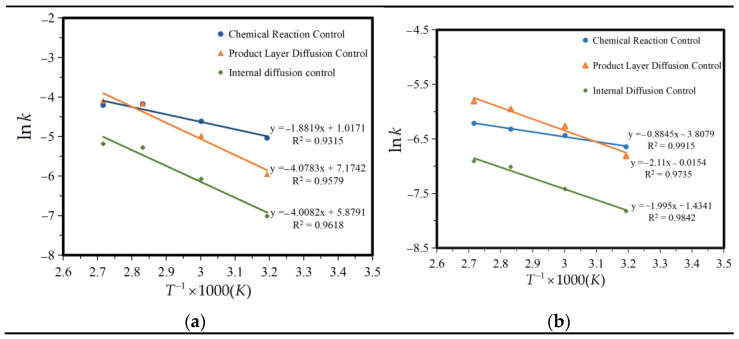
Fitting of ln *k* and 1/*T.* (**a**) 2–20 min, and (**b**) 20–90 min.

**Figure 10 molecules-28-00952-f010:**
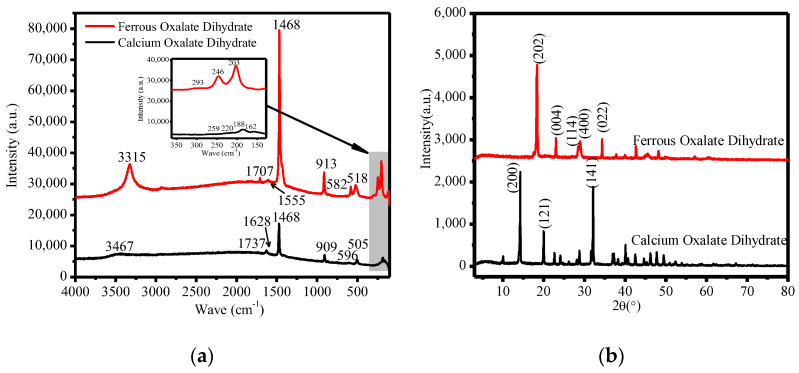
The filtrate residue of the leaching liquid treatment. (**a**) The Raman spectra of calcium oxalate and ferrous oxalate, and (**b**) the XRD diagram of calcium oxalate and ferrous oxalate.

**Table 1 molecules-28-00952-t001:** The experimental conditions.

Number of Samples	Single-Factor Experiment				
	Mass Fraction of Citric Acid *N* (%)	Reaction Temperature *T* (°C)	Liquid–Solid Ratio *S* (mL/2.5 g)	Reaction Time *t* (min)	Number of Cycles X
TiG-N	0, 1, 2.5, 5, 7.5, 10, 12.5	80	10	60	1
TiG-T	10	25, 40, 50, 60, 70, 80, 90, 95	10	60	1
TiG-*t*	10	40, 60, 80, 95	6, 7, 8, 9, 10, 11, 12,13, 14	10	1
TiG-S	10	80	8	10, 20, 30, 40, 50, 60, 70, 80, 90	1
TiG-X	10	80	8	90	1, 2, 3

**Table 2 molecules-28-00952-t002:** Chemical composition of the TiG Samples by XRF (wt %).

Chemical Composition	wt %	Chemical Composition	wt %
SO_3_	39.13	K_2_O	0.1
CaO	38.48	Cl	0.24
Fe_2_O_3_	11.18	As_2_O_3_	0.04
TiO_2_	2.52	SrO	0.04
SiO_2_	4.05	ZnO	0.03
MgO	2.08	ZrO_2_	0.01
Al_2_O_3_	1.16	P_2_O_5_	0.02
Na_2_O	0.62	loss of ignition	21.6
MnO	0.33		

**Table 3 molecules-28-00952-t003:** The radioactive test results of the TiG samples.

Test items		GB6656-2010					Test Results
Radioactivity		A	B		C		
	Internal exposure index	≤1	≤1.3	Does not satisfy Class A	-	Does not satisfy Class A or B	0
	External exposure index	≤1.3	≤1.9		≤2.8		0.1

**Table 4 molecules-28-00952-t004:** Chemical composition of the samples by XRF with different numbers of cycles (wt %).

TiG-X1		TiG-X2		TiG-X3	
Chemical Composition	wt %	Chemical Composition	wt %	Chemical Composition	wt %
SO_3_	50.56	SO_3_	33.02	SO_3_	34.43
CaO	41.07	CaO	57.10	CaO	42.49
Fe_2_O_3_	1.47	Fe_2_O_3_	1.92	Fe_2_O_3_	8.90
TiO_2_	1.26	TiO_2_	2.37	TiO_2_	2.95
SiO_2_	3.58	SiO_2_	3.64	SiO_2_	7.76
Al_2_O_3_	1.20	Al_2_O_3_	0.94	Al_2_O_3_	1.36
MgO	0.53	MgO	0.54	MgO	1.36
K_2_O	0.18	K_2_O	0.11	K_2_O	0.13
MnO	-	MnO	0.18	MnO	0.31
Na_2_O	0.08	Na_2_O	0.06	Na_2_O	0.07
ZnO	-	ZnO	-	ZnO	0.07
SrO	0.04	SrO	-	SrO	0.04
P_2_O_5_	0.03	P_2_O_5_	0.03	P_2_O_5_	0.03
ZrO_2_	0.01	ZrO_2_	0.02	ZrO_2_	0.02
Cl	-	Cl	-	Cl	0.02
As_2_O_3_	-	As_2_O_3_	-	As_2_O_3_	0.04
loss of ignition	19.1	loss of ignition	19.6	loss of ignition	20.9

‘-’ indicates that the detection limit was not reached.

**Table 5 molecules-28-00952-t005:** The apparent activation energy *E_a_* and frequency factor *k*_0_.

Time (min)	Control	*E*_a_ (kJ·mol^−1^)	Reference Range [36,37] (kJ·mol^−1^)	Match	*k*_0_ (S^−1^)
2–20 min	Chemical reaction control	15.65	>40	No	-
	Product layer diffusion control	33.91	20–40	Yes	1305.315
	Internal diffusion control	33.32	<20	No	-
20–90 min	Chemical reaction control	7.35	>40	No	-
	Product layer diffusion control	17.51	20–40	No	-
	Internal diffusion control	16.59	<20	Yes	0.238

## Data Availability

Not applicable.

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
