# Peer review of "The Leaching Kinetics of Iron from Titanium Gypsum in a Citric Acid Medium and Obtain Materials by Leaching Liquid"

_molecules, 2023, doi:10.3390/molecules28030952_

Round 1

Reviewer 1 Report

1. The title of the manuscript is inconsistent with the content of the manuscript, which needs to be revised.

2. State novelty and /or problem statement of the study from the Abstract. The abstract took some effort for me to understand. The graphical abstract is much better representation in this paper. Modify the abstract for clarity.

3. The novelty of this study could be more emphasized. Especially, the introduction section needs to re-organize. The major debate or Argument is not clear stated in the introduction session. Hence, the contribution debates are weak in this manuscript. I would suggest the author to enhance your literature discussion and arrives your debate or argument.

4. Section 2.1: The chemical composition, concentration (%) of the reagents should be provided.

5. The existence of iron in titanium gypsum requires additional characterization tests to verify.

6. While the results of leaching tests were discussed adequately, no information was given regarding the reproducibility of these test conditions.

7. Please make sure your conclusions' section underscores the scientific value-added of your paper, and/or the applicability of your findings/results. Highlight the novelty of your study. In addition to summarising the actions taken and results, please strengthen the explanation of their significance. It is recommended to use quantitativereasoning comparing with appropriate benchmarks, especially those stemming from previous work.

8. The figures and tables are suggested to be revised with higher quality. For example, delete unnecessary arrows, units.

Reviewer 2 Report

Titanium gypsum is a hard-to-treat solid waste. Small particles and high content of free water and ferric hydroxide with red color make titanium gypsum difficult to use in building materials. In this manuscript, the dissolution rule of ferric hydroxide of titanium gypsum are studied by leaching method using citric acid as the matrix and obtain high-value-added calcium oxalate and ferrous oxalate by leaching liquid. Before considering for publication, there are following suggestions for improving the manuscript quality.

1. Why do the amount of titanium gypsum dissolution increase with increased reaction temperature? 

2. Please confirm the sentence “at 80 °C and 60min, with an iron removal rate of up to 82.14% or 81.98% in the part of abstract”.

3. Please uniformly use “mass or quality ”in Figure 2-5, as well as in the text.

4. What is the optimized condition for leaching in the abstract?

5. All figures should be changed for better visibility.

Reviewer 3 Report

1. The tittle should be rewritten. It is very confusing. I think it should be ‘efficient removal iron from titanium gypsum’.

2. In Figure 2(b), the peaks of gypsum and calcium citrate tetrahydrate are very similar. How to distinguish them?

3. In Figures 2-6, there is no evidence for the evolution of iron-containing phases. So, it is difficult to analyze the removal mechanism of iron. Please explain it.

4. What are the values of pH of the leaching liquid before and after leaching?

5. In kinetics model, what is the difference between the product layer diffusion and internal diffusion. During leaching, what is the product layer in titanium gypsum? Whether the ‘internal diffusion’ is the iron diffusion through the titanium gypsum?

Round 2

Reviewer 3 Report

It can be accepted now.